# UAV Cluster Mission Planning Strategy for Area Coverage Tasks

**DOI:** 10.3390/s23229122

**Published:** 2023-11-11

**Authors:** Xiaohong Yan, Renwen Chen, Zihao Jiang

**Affiliations:** 1College of Aerospace Engineering, Nanjing University of Aeronautics and Astronautics, Nanjing 210016, China; rwchen@nuaa.edu.cn (R.C.); 2201013@nuaa.edu.cn (Z.J.); 2College of Artificial Intelligence, Xinjiang Vocational and Technical College of Communication, Urumqi 831401, China

**Keywords:** UAV, UAV cluster, task assignment, path planning, area coverage

## Abstract

In the context of area coverage tasks in three-dimensional space, unmanned aerial vehicle (UAV) clusters face challenges such as uneven task assignment, low task efficiency, and high energy consumption. This paper proposes an efficient mission planning strategy for UAV clusters in area coverage tasks. First, the area coverage search task is analyzed, and the coverage scheme of the task area is determined. Based on this, the cluster task area is divided into subareas. Then, for the UAV cluster task allocation problem, a step-by-step solution is proposed. Afterward, an improved fuzzy C-clustering algorithm is used to determine the UAV task area. Furthermore, an optimized particle swarm hybrid ant colony (PSOHAC) algorithm is proposed to plan the UAV cluster task path. Finally, the feasibility and superiority of the proposed scheme and improved algorithm are verified by simulation experiments. The simulation results show that the proposed method achieves full coverage of the task area and efficiently completes the task allocation of the UAV cluster. Compared with related comparison algorithms, the method proposed in this paper can achieve a maximum improvement of 21.9% in balanced energy consumption efficiency for UAV cluster task search planning, and the energy efficiency of the UAV cluster can be improved by up to 7.9%.

## 1. Introduction

In recent years, with the continuous development of unmanned aerial vehicle (UAV) technology, the application of UAVs in area coverage tasks such as plant protection, target search, and forest fire prevention has gradually become a reality [1,2,3,4,5]. UAVs have the advantages of high flexibility and task efficiency compared to traditional manual or mechanical operations in area coverage tasks [6,7]. Particularly in harsh environments, complex terrains, and areas with limited accessibility, using UAVs for search or agricultural operations can overcome environmental constraints, significantly reduce labor costs, and demonstrate the significant value of UAV applications [8,9]. In the process of using drones to perform tasks, a single UAV cannot meet the requirements of the task. Therefore, it is necessary to introduce multiple drones (referred to as a UAV cluster when small-scale drone operations occur or a UAV swarm when the number of drones increases) to meet the task requirements through task allocation, collaborative search, and situational awareness technologies.

Therefore, many experts and scholars have conducted research on UAV task assignments in area coverage. For example, Ref. [10] proposed an opposition-based learning parameter-adjusting harmony search algorithm to enhance the efficiency of UAV cluster task assignment. Ref. [11] presented a UAV task assignment method based on a simulated annealing strategy, which improved the task balance among UAV cluster individuals to a certain extent. Ref. [12] designed a distributed grouping cooperative dynamic task assignment method based on the extended contract net protocol, considering multiple constraints such as task writing, execution order, and environmental factors among UAV clusters, thereby optimizing the task execution efficiency. Ref. [13] addressed the problem of UAV trajectory optimization in three-dimensional complex environments and proposed a trajectory planning strategy based on an improved Harris Hawks algorithm. Ref. [14] focused on UAV participation in search and rescue scenarios and proposed an optimization strategy based on joint or decoupled optimization, which improved the search efficiency of UAVs.

We have summarized and categorized some existing research, and the results are shown in Table 1.

From Table 1, it can be seen that the aforementioned research mainly focuses on the efficiency of UAV cluster tasks, task load balance among UAV cluster individuals, and UAV trajectory planning. The methods used in these studies mainly involve mathematical modeling and optimization using swarm intelligence algorithms [18].

In addition, the experimental settings in the mentioned research are mostly in two-dimensional planes with small experimental scopes, and some studies involve only a single or a few UAVs. As research progresses and task scenarios expand, the study of UAVs in the field of area coverage tasks is becoming more diverse and complex, involving larger numbers and clusters of UAVs. In this context, the key focus of research is to design appropriate task allocation algorithms and achieve efficient task allocation and high-quality trajectory planning for UAV clusters based on relevant application scenarios [19,20].

Furthermore, the full coverage tasks of UAV clusters are mainly conducted in terrains with varying elevations, such as hills, mountains, and basins. UAV path planning needs to further consider different operating scenarios (altitude, wind speed, temperature, humidity, etc.) and make corresponding decisions for specific flight environments in order to adapt to different application situations [21]. Therefore, it is necessary to consider the impact of terrain variations on task allocation and trajectory planning for UAV clusters. Thus, it is essential to extend the research on efficient task allocation for UAV clusters in three-dimensional space beyond the two-dimensional plane.

As the task area increases and the computational complexity of three-dimensional scenarios grows, there is an urgent need to improve the efficiency of UAV cluster task planning algorithms [22]. In light of this, we first conduct a detailed analysis of area coverage tasks and provide further explanations of the research background. Then, we establish models for task area search, task area division, and UAV cluster cooperative task trajectory planning and derive the energy consumption model for UAV clusters. Next, we use an improved fuzzy C-means clustering method to divide the task area of the UAV cluster. Based on this, we propose a strategy that combines an optimized particle swarm algorithm with an ant colony algorithm for UAV cluster trajectory planning. Finally, the feasibility and effectiveness of the proposed approach are verified through relevant simulation experiments.

The schematic diagram of the proposed approach is illustrated in Figure 1.

The organization of this paper is as follows.

In Section 1, we provide a detailed analysis and summary of the research background and current status. In Section 2, we conduct a comprehensive analysis of area coverage tasks. In Section 3, we perform theoretical analysis and mathematical model derivation for UAV cluster task allocation and UAV cluster trajectory planning. In Section 4, we design relevant algorithms to solve the model optimization problems. In Section 5, we conduct simulation experiments and comparative experiments to evaluate the proposed strategies and discuss the results of these experiments. Finally, we summarize the research conducted in this paper and provide future prospects.

## 2. Overview of UAV Cluster Area Coverage Tasks

In this paper, area coverage tasks refer to tasks that require full coverage operations in a specific area, such as target search, forest fire inspection, and medication spraying in mountainous vegetation. Figure 2 illustrates the application scenarios.

This paper focuses on the research of rotary-wing UAVs. Compared to fixed-wing UAVs, rotary-wing UAVs have better maneuverability, making it easier to plan and allocate tasks for each UAV in the cluster. This enables faster and more optimal task allocation while ensuring algorithm efficiency and accuracy.

Before performing task allocation, we assume that the map of the task area has been constructed using satellite images or through UAV SLAM (simultaneous localization and mapping). Based on this, we establish a 3D coordinate system to accurately calibrate the relevant area, as shown in Figure 3.

## 3. Model Establishment

### 3.1. Coverage of UAV Search Area

In order to perform the area coverage task, UAVs are typically equipped with signal detection devices or actuators required for the task, such as infrared imaging devices, optical cameras, flame detection sensors, and spraying mechanisms. Taking infrared cameras and conventional onboard optical cameras as examples, these detection devices generally employ a cone-based detection method. The UAV serves as the vertex of the cone, and the target detection process is performed in a conical-shaped spatial area. For UAV cluster search tasks in the near-ground area, the UAV cluster carries search sensors, and the search model is illustrated in Figure 4.

Taking infrared area search as an example, the sensor carried by the UAV exhibits a conical beam search on the near-ground surface. The search radius and search coverage resolution vary at different heights. 

In the process of executing the search task with an infrared sensor, the search range and search resolution are the main technical performance indicators. Due to the attenuation of infrared radiation in the medium, the search range varies for different search targets and environments. In this study, based on actual conditions, the maximum searchable height for a single UAV carrying a sensor is determined.

As for the search resolution, it should meet the minimum resolution required for imaging the target under the condition of a limited imaging field of view. Considering both imaging resolution and imaging height, the maximum searchable distance satisfying these conditions is determined. The spatial search model for a single UAV is as follows: (1)$$h_{s}<h_{smax}$$
(2)$$S_{s}={\left(h_{s}\cdot tan\alpha_{s}\right)}^{2}\cdot pi$$
(3)$$R_{s}=\frac{{\left(h_{s}\cdot {tan\alpha }_{s}\right)}^{2}\cdot pi}{4}\cdot \frac{h_{s}}{2\cdot S_{s}}$$
where $h_{s}$ represents the actual height during the UAV search process, $h_{smax}$ represents the maximum value of UAV search height, indicating that the UAV cannot perform the search task when it exceeds this height; $S_{s}$ represents the area range that the UAV can project onto the ground from near-ground surface coverage at the current height; $\alpha_{s}$ represents the cone angle during the search process; $R_{s}$ represents the search resolution of the sensor carried by the UAV under the current height condition.

Using the conical area coverage method, the UAV cluster in Figure 5 can achieve full coverage of the task area. The coverage result is shown in Figure 5.

In Figure 5, the color depth represents the elevation of the terrain in the actual environment (darker color indicates lower altitude). During the UAV’s search for ground targets, an optimal solution for the ideal search height needs to be obtained based on the UAV’s current position.

### 3.2. Division of Task Area for UAV Cluster

In the process of UAV cluster operations, it is necessary to divide the task area into sub-areas. Unlike the uniform division of general task areas, UAVs mainly encounter the following three situations when searching for task points within the task area.

In an actual UAV near-ground search, the spatial relationship between the UAV and the near-ground search area is shown in Figure 6.

As shown in the figure above, α represents the search cone angle of the UAV. When conducting search tasks, the UAV’s coverage area may be on flat terrain. However, in most cases, the terrain is not flat but has certain height variations. For different near-ground search scenarios at different heights, two additional situations can be further distinguished, as illustrated in Figure 7.

In Figure 7, α represents the search cone angle of the UAV. $\beta_{1}$ and $\beta_{2}$ are the angles between the terrain and the horizontal plane in two different scenarios. In the top left image, $\beta_{1}$ < α, indicating that the UAV’s movement is similar to that on flat ground. In the top right image, $\beta_{1}$ > α; under the consideration of a safe distance for near-ground search, there exists a blind zone in the low-lying terrain where the UAV cannot perform the search. This is illustrated in Figure 8.

In the above figure, the brown-colored area represents the search blind zone for the UAV. In this case, the UAV needs to move directionally along the low-lying terrain to satisfy the search objectives in that area. This is illustrated in Figure 9.

For UAV cluster-based area coverage tasks, a reasonable partitioning of the task area can improve the operational efficiency of the UAV cluster and reduce search energy consumption.

Furthermore, during the process of UAV cluster-based area partitioning, since the comprehensive paths and energy consumption of the UAVs have not been fully obtained, it is necessary to initialize the task partitioning process based on ground reference points. These points should be selected as task points with a certain height that satisfies the UAV’s flight safety distance conditions.
(4)$$h_{sm}\leq h_{smax}$$
$h_{sm}$ is the height of the task points, and the maximum height is the searchable farthest altitude of the UAV. In this study, the impact of the partitioning results on the workload balance among individual UAVs within the UAV cluster is considered as the evaluation criterion for the UAV cluster partitioning method.

The process of establishing the balance model for UAV search area partitioning in this study is as follows:(5)$${h_{sdis}=h}_{smax}-h_{smin}$$
(6)$$h_{sav}=(\sum h_{s})/n_{s}$$
(7)$$x_{sav}=(\sum x_{s})/n_{s}$$
(8)$$y_{sav}=(\sum y_{s}i)/n_{s}$$
where $h_{smax}$ represents the maximum height within the current UAV partitioned area; $h_{smin}$ represents the minimum height within the current UAV partitioned area; $h_{sdis}$ represents the maximum height difference within the current UAV partitioned area. $h_{s}$ represents the height of the search task points that the UAV needs to pass through; $n_{s}$ represents the number of task points that the UAV needs to pass through within its assigned area; $h_{sav}$ represents the average task operation height of the UAV within its assigned area. $x_{s}$ represents the sum of the x-coordinate values of all the points that the UAV needs to pass through; $x_{sav}$ represents the average x-coordinate value of the points within the spatial range that the UAV needs to pass through. $y_{s}$ represents the sum of the y-coordinate values of all the points that the UAV needs to pass through; $y_{sav}$ represents the average y-coordinate value of the points within the spatial range that the UAV needs to pass through. 

The definition of the workload difference between individual UAVs within the UAV cluster can be expressed as follows:(9)$$V_{s,s+1}=\frac{\begin{array}{c} \sqrt[{2}]{\sum_{i=1}^{n_{s}}{(h}_{s+1}-h_{\left(s+1\right)av})-\sum_{i=1}^{n_{s}}{(h}_{s}-h_{sav})}+\\ \sqrt[{2}]{\sum_{i=1}^{n_{s}}{(x}_{s+1}-x_{\left(s+1\right)av})-\sum_{i=1}^{n_{s}}{(x}_{s}-x_{sav})}+\\ \sqrt[{2}]{\sum_{i=1}^{n_{s}}{(y}_{s+1}-y_{\left(s+1\right)av})-\sum_{i=1}^{n_{s}}{(y}_{s}-y_{sav})} \end{array}}{{(D}_{s+1}-D_{s})}$$
In the above equation, *V*_*s,s*+1_ represents the difference in task volume between different UAV individuals, *D_s_* is the search traversal distance value for UAV individuals. After defining the workload difference model, the task allocation in the cluster is based on the workload difference among UAVs within the task area.

### 3.3. UAV Path Planning

The UAV cluster path planning model involves deriving a mathematical model for UAV task path planning based on the determined task areas for individual UAVs.

First, the path motion of a single UAV is illustrated in Figure 10. 

In Figure 10, the solid line arrow represents the direction of UAV search movement, while the dashed line arrow represents the actual movement process. During the search process, a single UAV should cover the area in a circular pattern, while the arrangement of the required task points follows a hexagonal pattern. As a result, there will be overlapping search areas during the UAV search process. In order to minimize the overlap, the UAV’s movement direction is illustrated in Figure 11.

In Figure 11, the solid line represents the direction of the UAV’s search movement, while the dashed line represents the actual trajectory. In the search process, a single UAV should ideally cover a circular area, but the arrangement of task points follows a hexagonal pattern, resulting in overlapping search areas. To optimize the search process in the task area, the UAV’s movement direction should be such that it minimizes the repeated coverage area when reaching each task point. However, in practical planning, the focus is on achieving the overall shortest path, and it may not be possible to eliminate the UAV’s actual trajectory. This study introduces an optimization condition that minimizes the repeated coverage area when each task point is reached while considering the global path optimization based on the shortest path algorithm. The specific research process for UAV’s trajectory planning model is described as follows:

Based on the criteria of achieving the fastest full coverage of task points within the area, the definition of search efficiency is as follows:(10)$$t_{s}=\sum_{i=1}^{k_{m}}\frac{{|p}_{s}-p_{s+1}|}{v_{s}}$$
where $p_{s}$ represents the current position of the UAV; $p_{s+1}$ represents the position of the next task point that the UAV needs to operate on; $k_{m}$ represents the number of task points within the UAV’s allocated area; $v_{s}$ represents the search speed of the UAV. $t_{s}$ represents the total time required for the UAV to complete the full search of the task points within its allocated area.

When a UAV completes all search tasks within a task area of size A in time *t*, the search efficiency can be defined as
(11)$$e_{fs}=\frac{S_{s}}{t_{s}}$$
where $S_{s}$ represents the total area of UAV search, and $e_{fs}$ represents the efficiency value based on the fastest trajectory planning objective. 

The search model based on minimizing the repeated area when arriving at new task points can be defined as
(12)$$S_{rs}=\sum_{i=0}^{k}\left(S_{fs}-S_{cs}\right)$$
where $S_{fs}$ is the actual search area of the UAV; $S_{cs}$ is the theoretical coverage area of UAV task points; $S_{rs}$ is the area of repeated search generated by the UAV; and k is the number of task points already completed by the UAV. 

Based on minimizing the repeated area when arriving at task points, the efficiency definition can be expressed as
(13)$$e_{as}=\frac{S_{rs}}{t_{s}}\times \beta_{a}$$
where $e_{as}$ represents search efficiency for the lowest repeat area objective; $\beta_{a}\;$ represents a constant coefficient that provides an equivalent calculation for two different search efficiencies. By establishing models for two search efficiencies in the UAV cluster’s trajectory planning process, the search tasks of the UAV cluster can be carried out with a balanced and efficient search for the search objectives and task areas as much as possible.

### 3.4. Energy Efficiency Model for the UAV Cluster

During the process of conducting area coverage search tasks, energy consumption is an important optimization criterion for the UAV cluster in addition to achieving the objective of complete coverage. Adopting a low-power search approach enables the UAV cluster to have a larger operational range during task execution. Additionally, the reduced power consumption during movement can compensate for the energy usage of the UAV’s onboard sensors, further improving search efficiency.

The energy consumption model is defined in the following process.

First, for a single UAV in the spatial task area, the distance traveled is represented by l, the length of the route, and the time spent on traveling is represented by t. The energy consumption is calculated as follows without changing the direction of movement.(14)$$E_{Sl}={l_{sl}\cdot E}_{il}+{{l_{sv}\cdot E}_{iv}+t\cdot E}_{b}$$
where $E_{il}$ represents the energy consumption per unit distance in the horizontal direction for the UAV, $E_{iv}$ represents the energy consumption per unit distance in the vertical direction, $l_{sl}$ represents the position change in the horizontal direction, $l_{sv}$ represents the position change in the horizontal direction, and $E_{b}$ represents the energy consumption during the idle time while the UAV is in motion. 

At the UAV’s turning points, we define the energy consumption for all turns as $E_{Sd}$, meaning that each turning event incurs an energy cost of ${\;E}_{Sd}$.

For a single UAV operating within its assigned task area with a known number of turns *k**_sd_* and the number of task points k, according to the trajectory planning algorithm, the distance the UAV needs to fly is $l_{s}$. The energy consumption incurred by a single UAV in completing the full coverage search task within the assigned area is given by $E_{Su}$.
(15)$$E_{Su}=\sum {l_{sl}\cdot E}_{il}+\sum {l_{sv}\cdot E}_{iv}+{t\cdot E}_{b}+k_{sd}\cdot E_{Sd}$$

Furthermore, assuming that the total number of task points in the search area required by the UAV cluster is K, and the number of UAVs in the cluster is n, we have
(16)$$K=k_{1}+k_{2}+k_{3}+\ldots +k_{n}$$

Then, by using the trajectory planning algorithm, we can obtain the flight paths required for UAVs to search within their respective areas, denoted as $l_{1},l_{2}\ldots l_{n}$. Therefore, under the current area partitioning method and trajectory planning algorithm, the total power consumption for the UAV cluster to complete the area coverage search in the task area is given by
(17)$$E_{Su}=\sum_{i=1}^{n}l_{il}{\cdot E}_{il}+\sum_{i=1}^{n}l_{iv}{\cdot E}_{iv}+{t\cdot n\cdot E}_{b}+K\cdot E_{Sd}$$

Finally, based on the energy consumption of the UAV cluster during the search in the target area, we can further establish a model for the energy efficiency ratio. That is, based on the known energy consumption, we can establish the UAV cluster’s energy efficiency model in terms of the time required for the UAV search. 

Let the energy consumption of a single UAV to complete the search in its respective area be denoted as $E_{Sui}$, the duration of the search consumption be denoted as $t_{i}$. The energy efficiency of a single UAV is given by
(18)$$e_{i}=\frac{E_{Sui}}{t_{i}}$$

For the UAV cluster, energy efficiency can be divided into balanced energy efficiency and overall energy efficiency. Balanced energy efficiency is used to characterize the differences in energy efficiency among UAVs within the cluster. Overall energy efficiency, on the other hand, compares and verifies the energy efficiency of the UAV cluster as a whole.

In the UAV cluster, the individual energy consumption of each UAV is given by $E_{Su1},\;E_{Su2},\;E_{Su3},\;\ldots ,\;E_{Sun}$, and the corresponding time consumed by each UAV is given by $t_{1},\;t_{2},\;t_{3},\;\ldots ,\;t_{n}$. The model for balanced energy efficiency is as follows
(19)$$E_{Sua}=\sum_{i=1}^{n}\frac{E_{Sui}}{n}$$
(20)$$e_{b}=1-\sum_{i=1}^{n}\frac{E_{Sui}-E_{Sua}}{t_{i}}/a_{b}$$
where $a_{b}$ is constant for balanced energy efficiency.

The total energy efficiency model is as follows
(21)$$e_{a}=\frac{\sum_{i=1}^{n}E_{Sui}}{\sum_{i=1}^{n}t_{i}}$$

By introducing the overall energy efficiency of the UAV cluster and the energy efficiency of individual UAVs, the algorithm performance can be comprehensively evaluated.

## 4. UAV Cluster Search Task Algorithm Design

### 4.1. UAV Cluster Task Area Planning

The UAV cluster search task algorithm mainly consists of the search task area division of the UAV cluster and the trajectory planning algorithm design for UAVs within the corresponding areas. 

The task area division of the UAV cluster is aimed at achieving the fastest coverage of the specified area under the assumption of the same search efficiency and motion state for multiple UAVs. Traditional clustering methods have difficulty in clearly clustering edge points and rely solely on the number of iterations to determine the completion of clustering tasks [23]. To address this issue, this paper proposes an improved fuzzy C-means clustering method (O-FCM) for the task area division of the UAV cluster. 

The convergence process of the cluster centers is as follows:(1)In this algorithm, the membership function is modified by incorporating the Euclidean distance between the current traversed node and its adjacent nodes as the relevance weight. This modification enhances the similarity of node classification. The optimized membership function is as follows:
(22)$$u_{ij}=\frac{1}{{{(d\sum_{k=1}^{c}{\left(\frac{d_{ij}}{d_{ik}}\right)}^{\frac{2}{m-1}}}_{imin})}^{\sigma }}$$
where $d_{imin}$ represents the Euclidean distance to the nearest node from the current node, and σ represents the inclusion of the influence factor of the nearest distance in determining the weight of the membership function. Regarding the method for selecting the nearest node, instead of calculating the Euclidean distance between the current node and all other nodes in the area, the algorithm sorts all nodes in the area in ascending order based on their positions along the X, Y, and Z axes. It then calculates the Euclidean distance between the current node and the nearest node along each axis (X, Y, and Z) separately. Finally, it selects the minimum value among the three distances as the Euclidean distance between the nearest node and the current node.

(2)At this point, the calculation of the new centroids within the current category is performed based on the following steps:


(23)
$$c_{i}=\frac{\sum_{i=1}^{n}u_{ij}^{m}{\vec{x}}_{i}}{\sum_{i=1}^{n}u_{ij}^{m}}$$


Then, based on the membership degrees and coordinate values of each node within the area, the optimal coordinates of the current clustering center nodes within the area can be obtained.

(3)The objective function value under the current condition is calculated as follows:


(24)
$$J_{z}=\sum_{i=1}^{c}J_{i}=\sum_{i=1}^{c}\sum_{j}^{n}u_{ij}^{m}d_{ij}^{2}$$


Further, the convergence condition of the current clustering algorithm is determined based on Equation (25).
(25)$$J_{z+1}-J_{z}<\eta_{J}$$
when the convergence of the objective function reaches an accuracy less than $\eta_{J}$, it can be determined that the current clustering algorithm satisfies the termination condition.

In this study, for the near-ground UAV area in space, the search coverage is a planar coverage method, which can be further equivalently represented as circular coverage of the planar area. The overlapping area around the circular area cannot be directly calculated to obtain the distance between the centers of adjacent circles in practical situations. Therefore, this study uses polygons as a replacement for circles to achieve the minimal overlapping area as much as possible. The relevant calculations are as follows:(26)$$\left\{\begin{array}{c} \eta =180-\frac{360}{k_{s}}\\ k_{s}*\partial_{s}=360 \end{array}\right.$$
where $k_{s}\;$ represents the number of sides of the polygon, and $\partial_{s}$ represents the number of internal angles of the polygon. Since the number of sides of a polygon should be a positive integer, solving the equation simultaneously yields a hexagonal polygon with the maximum number of sides, which corresponds to the minimum overlapping area.

By constructing a honeycomb-like continuous hexagonal coverage area, the search range of the UAV cluster is obtained. The honeycomb pattern formed by hexagons represents the actual movement coverage range for the UAVs. The common center of the circles and hexagons represents the coordinates that the UAVs need to pass through.

Here is the pseudocode for the O-FCM algorithm (See Algorithm 1):

**Algorithm 1.** O-FCM1:**Start the task**.2:**Initialization:** Initialize the clustering iteration parameter3:Randomly generate cluster centers.4:
**Repeat:**
5: Recalculate the cluster center based on the current category6: Update membership function based on current clustering7: Regenerate the cluster based on the current cluster8:
**Until:**
9: The objective function satisfies the convergence condition10:Get the cluster center and the categories that each node belongs to11:
**End**


### 4.2. UAV Trajectory Planning

The design of the UAV cluster trajectory planning algorithm is mainly based on the collaborative mission objectives of the UAV cluster, aiming to find a comprehensive solution for UAVs’ traveling process under the given environmental background and optimization conditions. To address the drawbacks of poor convergence and susceptibility to local optima in traditional swarm intelligence algorithms, this paper proposes an optimized particle swarm optimization hybrid ant colony algorithm (PSOHAC) to efficiently achieve trajectory planning for UAV clusters.

The ant colony algorithm primarily optimizes the model based on pheromone trails along the ant paths. However, the ant colony algorithm suffers from slow convergence speed [24]. In this study, we incorporate a dynamic pheromone calculation mechanism and a pheromone trail filtering mechanism into the original ant colony algorithm. The dynamic pheromone calculation mechanism uses a dynamic weight accumulation method to accumulate pheromone trails based on the existing pheromone accumulation. The number of accumulations is related to the weight. The pheromone trail filtering mechanism eliminates paths directly when the pheromone intensity on a path is smaller than the mean of all pheromone intensities and a fixed constant during the ant colony iteration. By incorporating these two mechanisms, the convergence speed is improved while ensuring the robustness and distributed nature of the ant colony algorithm.

In this article, the individual trajectory planning of UAVs from the starting point to the destination adopts the ant colony algorithm with dynamic pheromone weights. The algorithm is outlined as follows.

In the algorithm, the ant population is set to a total of ants $n_{ac}$. At each time step $t_{ac}$, the path taken by an individual ant is denoted as $l_{ij}$. After each ant passes through the path, if the path successfully leads to the destination, the intensity of pheromone on that path is
(27)$$\tau_{ij}\left(t_{k}\right)=\sqrt[{\beta_{ac}}]{{\alpha_{ac}\cdot n}_{ac}+\tau_{ij}\left(t_{k-1}\right)}$$
where $\tau_{ij}\left(t_{k}\right)$ represents the concentration of pheromone on the path from i to j at time $t_{k}$. $n_{ac}$ represents the number of times the current path has been traversed by ants. $\alpha_{ac}$ is a constant factor used in the adaptation of the ant colony algorithm’s pheromone calculation. $\tau_{ij}\left(t_{k-1}\right)$ represents the concentration of pheromone at the previous time step. $\beta_{ac}$ is a balancing constant coefficient used to ensure the correlation between the number of times and the concentration of pheromone, while avoiding getting trapped in local optima. 

The probability expression for the ant at position i to select path j in the algorithm is as follows:(28)$$p_{ij}^{k}=\left\{\begin{array}{c} \frac{\tau_{ij}\left(t_{k}\right)\cdot \eta_{ij}\left(t_{k}\right)}{\sum_{s\in all}\tau_{is}\left(t_{k}\right)\cdot \eta_{is}\left(t_{k}\right)},j\in all\\ 0,\text{others} \end{array}\right.$$
where all represents the set of all paths that can be reached from i, the current position of the ant. $\eta_{ij}\left(t_{k}\right)$ represents the visibility of the path from i  to j, which is related to the distance from the selected path to the destination.

Furthermore, the concentration of pheromone on the paths is updated after each generation of ants completes the journey from the starting point to the destination. The update equation is as follows:(29)$$\tau_{ij}\left(t+k\right)={\rho \cdot \tau }_{ij}\left(t\right)+∆\tau_{ij}$$
(30)$$∆\tau_{ij}\left(t+k\right)=\sum_{n_{i}=1}^{n_{ac}}∆\tau_{ij}^{n_{i}}$$
where ρ represents the persistence of the current pheromone on the path; $\tau_{ij}\left(t+k\right)$ is the updating process of pheromone concentration on the path at time k; $n_{ac}$ represents the total number of ants. 

During each iteration of the ant colony, the individual ants optimize the paths and evaluate their quality based on the pheromone concentration. In the next iteration of the ant colony, further iterative calculations are performed on the high-quality paths. After completing one iteration of the ant colony optimization, the existing pheromone concentration in the optimized paths is updated, and the iteration process continues in the next generation. After completing the trajectory optimization process for all individual ants and reaching the maximum iteration limit, the algorithm concludes the optimization process of UAV individual paths. The use of dynamic pheromone concentration improves the optimization efficiency of the ant colony algorithm by reducing the computational dataset under the condition of increased computational complexity. 

In the particle swarm optimization (PSO) algorithm, in each iteration, the best individual of the current generation is obtained by calculating the fitness value of all individuals. However, within each generation, the individuals have a certain degree of homogeneity in terms of fitness value and can be classified into different ranges of fitness values [25]. In this paper, the characteristics of the niche population mechanism are utilized to categorize the particle swarm population individuals in each generation [26]. Different dynamic weight proportions are assigned to the fitness values of different populations in each iteration, enhancing the comparability of the iteration of the internal groups within the particle swarm population. The specific content of the optimized particle swarm optimization algorithm is as follows.

(1)Based on the UAV cluster collaborative task model, the updating equation of the niche particle swarm algorithm is expressed as follows:(31)$$\left\{\begin{array}{c} V_{in}=\omega V_{in}+c_{1}r_{1}(p_{in}-x_{in})\\ +n_{mu}c_{2}r_{2}(p_{gn}-x_{in})\\ x_{in}=x_{in}+V_{in} \end{array}\right.$$
where $V_{in}$ represents the real-time velocity of a particle in the niche-optimized particle swarm algorithm, $x_{in}$ represents the current spatial position of the particle individual. $p_{in}$ represents the global best position, $p_{gn}\;$ is the best position among all positions traversed by all particle individuals. $c_{1},\;r_{1},c_{2},\;r_{2}$ are weight constants. h is the segmentation criterion for the niche population.

(2)After establishing the state update process for particle individuals, it is necessary to define the objective functions for particle individual optimization, niche population optimization, and the overall algorithm objective achieved by the final iteration of the particle swarm population. The objective function for particle individual optimization is defined as follows:

(32)$$(F_{pi}=C_{1}+f_{pi})|\sum_{i=1}^{n}f_{pi}\leq K$$
where $F_{pi}\;$ represents fitness function values representing individual particles; K represents the shortest spatial path achievable by a UAV cluster regardless of obstacles; $C_{1}$ represents the weighted weights of the small habitat population to which the particle fitness value belongs; $f_{pi}\;$ represents the current fitness value of the particle in the particle population.

(3)After achieving the optimization objectives and requirements for individual particles within the particle population, the process of dividing the particle population into small habitat populations is performed. In the small habitat particle swarm algorithm, the division rule for the small habitat population is determined based on the similarity of fitness value distribution in each generation of the particle population, as shown in the following equation:

(33)$$\left\{\begin{array}{c} n_{su},\;\;\;F_{pi}>\frac{\sum F_{pi}}{\left(k-i\right)}\cdot max\left(F_{pi}^{n}\right)\\ n_{mu},\;\;\frac{\sum F_{pi}}{\left(k-i\right)}\cdot min\left(F_{pi}^{n}\right)\leq F_{pi}\leq \frac{\sum F_{pi}}{\left(k-i\right)}\cdot max\left(F_{pi}^{n}\right)\\ n_{lu},\;\;F_{pi}^{n}<\frac{\sum F_{pi}}{\left(k-i\right)}\cdot min\left(F_{pi}^{n}\right) \end{array}\right.$$
where n is the number of individuals in the current particle population, i is the specific iteration count, k is the total number of iterations, $n_{su}$ is the set of excellent individuals, $n_{mu}$ is the set of balanced individuals, and $n_{lu}$ is the set of poor individuals. In the initial iterations, the screening degree of individual excellence is higher, and the number of individuals screened out is larger. As the iterations progress, the number of screened individuals decreases. Towards the end of the iteration, individuals are no longer eliminated.

(4)After determining the optimization objectives for individual particles within the particle population and the small habitat populations, the iterative optimization process begins. It involves the dynamic data link with the ant colony algorithm, enabling the two algorithms to collaborate in the optimization process. The small habitat particle swarm algorithm ultimately determines the final optimization result based on the number of iterations. Once the specified number of iterations is reached, the algorithm terminates and outputs the computed result.

The pseudocode for the PSOHAC algorithm is as follows (see Algorithm 2):

**Algorithm 2.** PSOHAC1:**Start the task**.2:**Initialization:** Initialize individual particle data, particle swarm fitness function, niche population parameters, particle population data and ant population parameters.3:
**Repeat-1:**
4:    Updated the ant sport count.5:    Calculate the state transition probability.6:    Computational analysis of pheromone.7:    Completion meets the iteration limit.8:    Update the particle population properties9:    **Repeat-2:**10:        Fitness function algorithm optimization.11:        **Repeat-3:**12:            Update individual fitness of particles.13:        **Until-3:**14:            Reach the limit of quantity.15:    **Until-2:**16:        Meet the iteration limit17:        Iterate the optimal cluster path.18:        Update pheromone properties19:
**Until-1:**
20:    Iterative optimal path.21:    Obtain optimal UAV cluster path planning.22:
**End**


## 5. Simulation Analysis

In this study, an optimized fuzzy C-means clustering algorithm and an optimized particle swarm algorithm combined with the ant colony algorithm are used to achieve task allocation for full coverage search of UAV clusters in a given area. The specific simulation parameter settings and experimental process are as follows. 

### 5.1. Algorithm Parameter Settings 

The parameters used in the algorithm are set as Table 2.

### 5.2. Simulation Experiment Testing

The simulation in this study is conducted in a horizontal range of 1 km × 1 km and a height range of 0 to 100 m. A 3D mountain model is generated within this range to represent the terrain. The specific illustration is presented in Section 2, Figure 3b.

First, based on the task points within the environment, simulation tests are conducted to evaluate the number of UAVs required for a complete search of the designated area.

In the task area, a higher number of UAVs will result in faster search times. However, in terms of energy consumption, the relationship between the number of UAVs and the energy consumed during the search is not straightforward. In this study, simulations are conducted to analyze the energy efficiency generated by different numbers of UAVs during the search operation. Multiple experiments are conducted under consistent conditions in terms of other parameters and derivation processes, and the average values are obtained. The test results are presented in the Figure 12.

According to Figure 12, the optimal energy consumption efficiency is achieved when the number of UAVs is eight.

In the aforementioned experimental environment, the improved fuzzy C-means clustering algorithm described in this study was applied. The spatial clustering results obtained in the mountainous planning area are depicted in Figure 13.

In the above figure, differently colored points represent the task points assigned to different UAVs. From the figure, it can be observed that the method described in this study has successfully accomplished the task allocation for the UAV cluster within the designated area. At this stage, the task area range corresponding to each UAV has been preliminarily identified. The next step is to further plan the UAV trajectories.

Using the PSOHAC algorithm described in this study, the task trajectory planning for the UAV cluster corresponding to the task area is performed. The results are shown in Figure 14.

From Figure 14, it can be observed that the PSOHAC algorithm described in this study has successfully performed trajectory planning for the respective task points of the eight UAVs. Each trajectory is smooth and without any intersections.

Furthermore, the fitness values for each of the eight UAVs are obtained, as shown in Figure 15. In the graph, the maximum difference in fitness values between UAVs is 14%, with an average difference of 9%. These results demonstrate the feasibility and effectiveness of the proposed approach described in this study.

### 5.3. Comparative Experimental Testing

To demonstrate the superiority of the improved algorithms proposed in this study, we conducted comparative experimental testing against several classical traditional algorithms under consistent parameter settings. In terms of task area partitioning, we compared the K-means clustering algorithm, the C-means clustering algorithm, and the fuzzy C-means clustering algorithm used in this study. For UAV cluster trajectory planning, we conducted comparative testing against the immune algorithm (GA) [27], ant colony optimization (ACO), particle swarm optimization (PSO) [28], and the simulated annealing (SA) algorithm [29].

First, to verify the performance of the PSOHAC algorithm, we conducted comparative experiments using the UAV trajectory planning model described in Section 3.3 as the test function. The relevant results are shown in Figure 16 and Figure 17.

Based on the comprehensive analysis of Figure 16 and Figure 17, it can be observed that the proposed PSOHAC algorithm has shown advantages in terms of convergence speed and optimization accuracy compared to the other three comparative algorithms. The dynamic pheromone calculation mechanism introduced in the ant colony algorithm, as well as the iterative process based on the small habitat theory in the particle swarm algorithm, have contributed to a 22% faster convergence speed and a 20% higher optimization accuracy.

Building upon these findings, consistent with the testing experiments in Section 5.2, further comparative experiments were conducted. The results of the comparative experimental testing are presented in the following Table 3. 

From Table 3, it can be observed that our proposed method exhibits higher balanced energy consumption efficiency compared to other comparative algorithms. In other words, in the process of UAV cluster task allocation, our method achieves a more balanced distribution of tasks among individual UAVs. In terms of task allocation, our method results in a lower total task quantity compared to other comparative algorithms. Furthermore, in terms of task efficiency, our method effectively reduces the length of UAV trajectories while ensuring full coverage of the task area.

## 6. Conclusions

This paper addresses the issues of high-energy consumption, uneven task allocation, and low task efficiency in unmanned aerial vehicle (UAV) cluster missions for area coverage. It proposes approaches from the perspectives of task area division and UAV trajectory planning.

Pre-processing of the task area is conducted, and cluster task planning models, UAV trajectory planning models, and UAV energy efficiency models are established. For the task area division problem in UAV cluster missions, an improved fuzzy C-clustering method is proposed. It incorporates neighborhood attributes in addition to node clustering attributes to assist in accurate node clustering.

For UAV trajectory planning, a strategy combining the optimization of particle swarm optimization with ant colony optimization is proposed; the strategy focuses on multi-objective optimization around minimizing repeated area and achieving faster global planning. The strategy aims to overcome the weaknesses of classical swarm intelligence algorithms, such as poor convergence and getting trapped in local optima.

The simulation experiment results demonstrate that the proposed strategies successfully accomplish task allocation and trajectory planning for UAV clusters within the designated task area. Comparative experiments show that the proposed algorithms exhibit improvements in optimization accuracy. Additionally, the trajectory planning strategy achieves advantages in terms of convergence speed.

Compared to other methods, the proposed approach can achieve a maximum improvement of 21.9% in balanced energy consumption efficiency for UAV cluster task search planning. The overall energy efficiency of the UAV cluster can be improved by up to 7.9%. The application of cluster task allocation and path planning presented in this paper has valuable reference value. It is worth noting that the simulations may have some deviations from real-world scenarios, and further adjustments to the parameters will be made through real-world testing in future research.

## Figures and Tables

**Figure 1 sensors-23-09122-f001:**
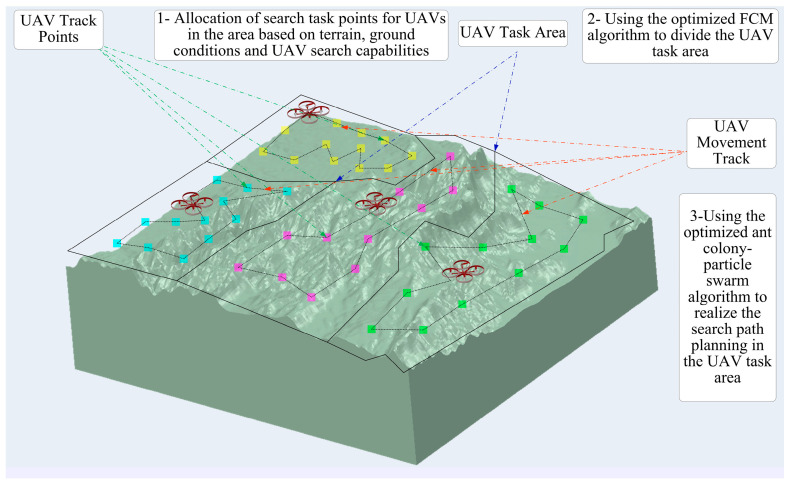
Schematic Diagram of the Proposed Approach.

**Figure 2 sensors-23-09122-f002:**
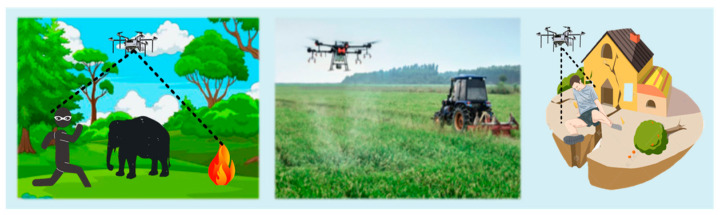
Schematic Diagram of Area Coverage Task Scenarios.

**Figure 3 sensors-23-09122-f003:**
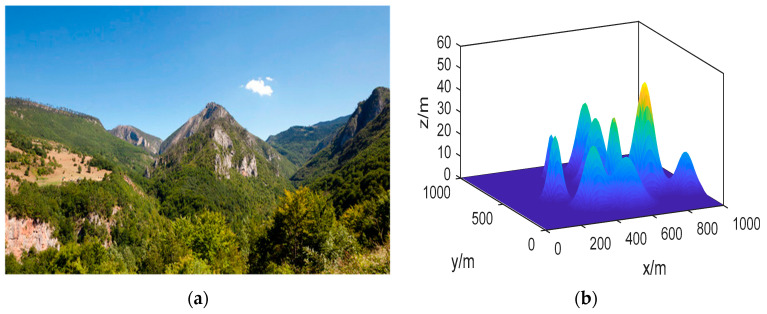
Illustration of Task Area Transformation and Calibration. (**a**) Actual Topographic Map; (**b**) Transformed Task Map.

**Figure 4 sensors-23-09122-f004:**
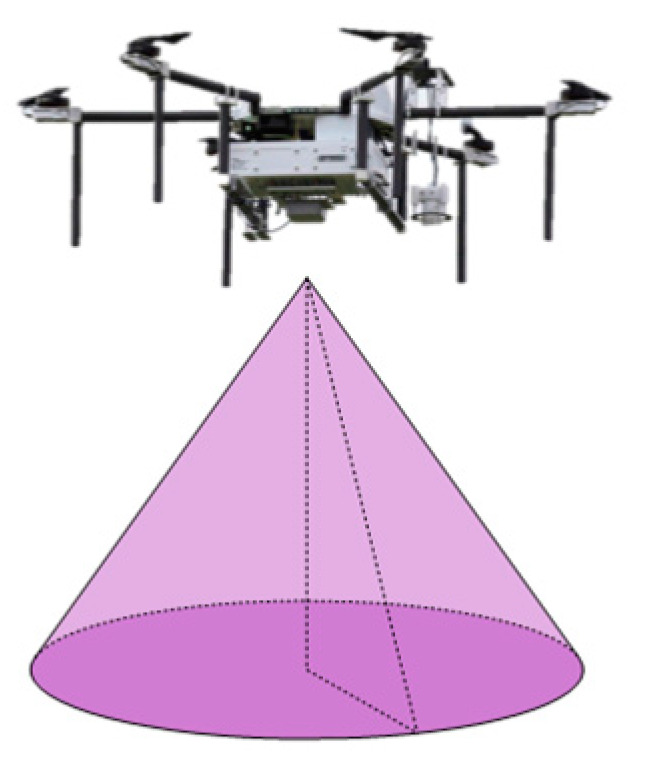
UAV Search Coverage Model.

**Figure 5 sensors-23-09122-f005:**
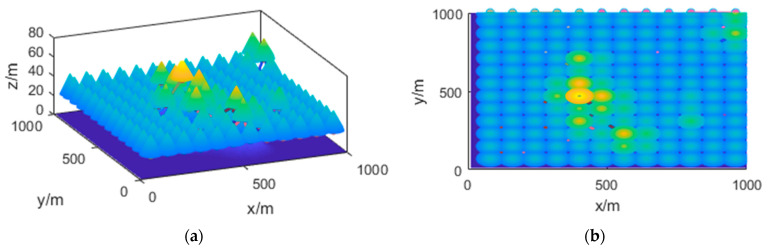
Task Area Coverage with UAV Cone Beam. (**a**) 3D view; (**b**) 2D view.

**Figure 6 sensors-23-09122-f006:**
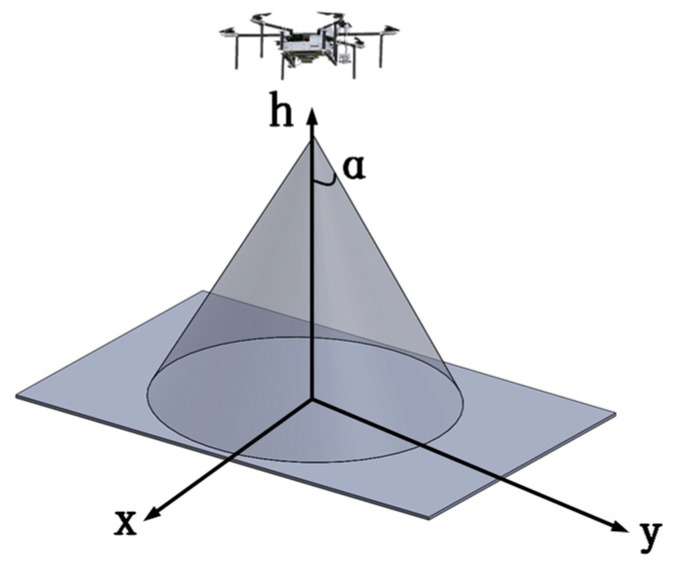
UAV Search Coverage Area on Flat Ground.

**Figure 7 sensors-23-09122-f007:**
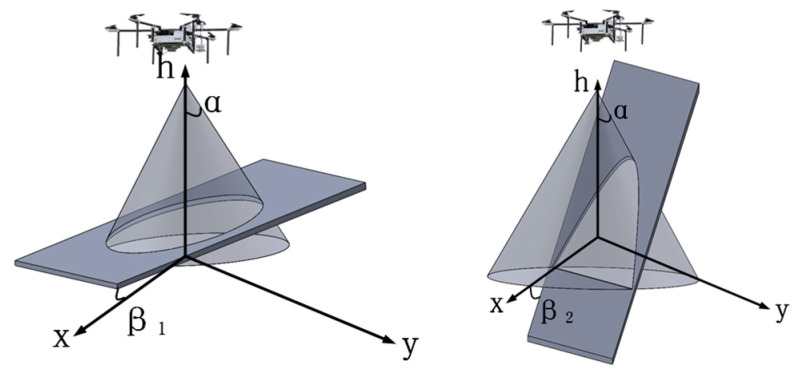
UAV Search Coverage Area on Sloped Terrain.

**Figure 8 sensors-23-09122-f008:**
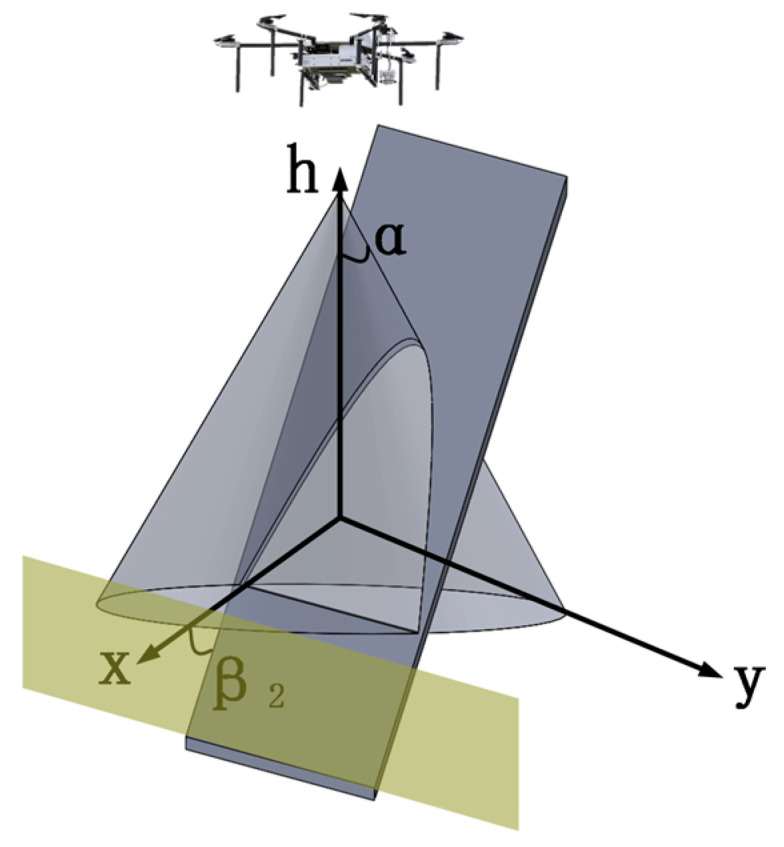
Search Blind Zone on Steep Slope with Incline Angle Greater than UAV Search Cone Angle.

**Figure 9 sensors-23-09122-f009:**
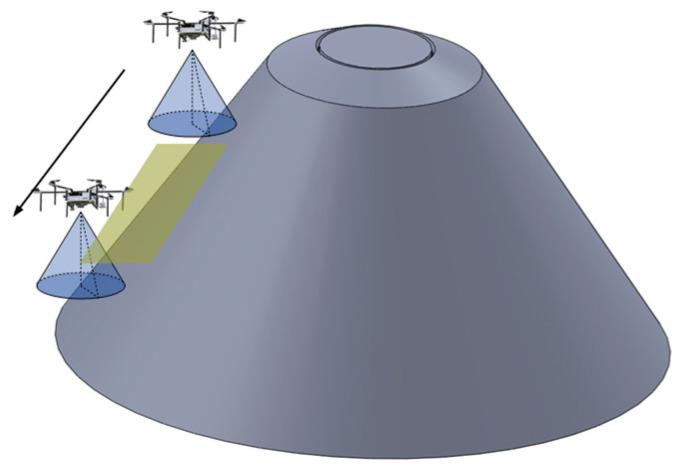
Directional Search of UAV on Steep Slope.

**Figure 10 sensors-23-09122-f010:**
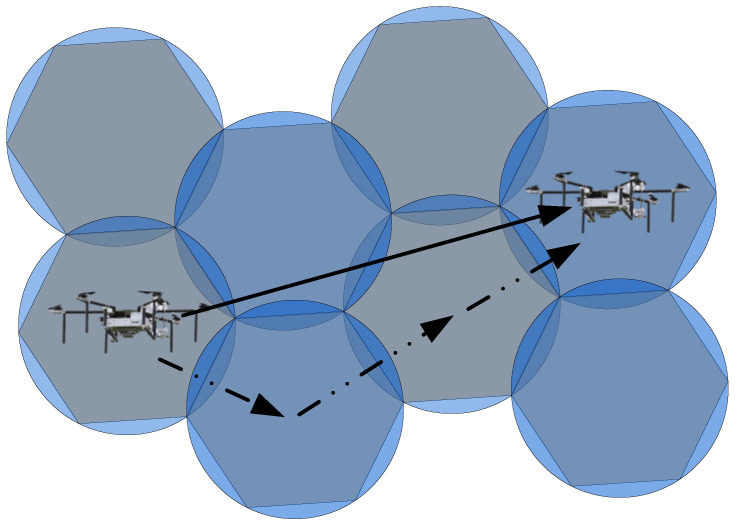
Path Motion Process of a Single UAV-1.

**Figure 11 sensors-23-09122-f011:**
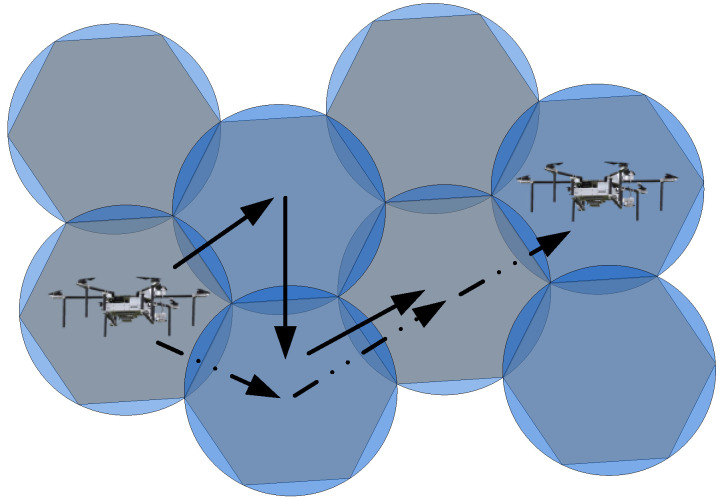
Path Motion Process of a Single UAV-2.

**Figure 12 sensors-23-09122-f012:**
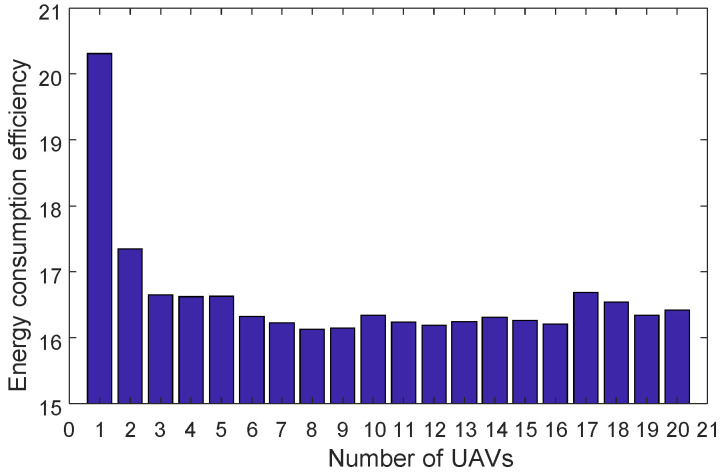
Relationship Between the Number of UAVs and Energy Consumption Efficiency.

**Figure 13 sensors-23-09122-f013:**
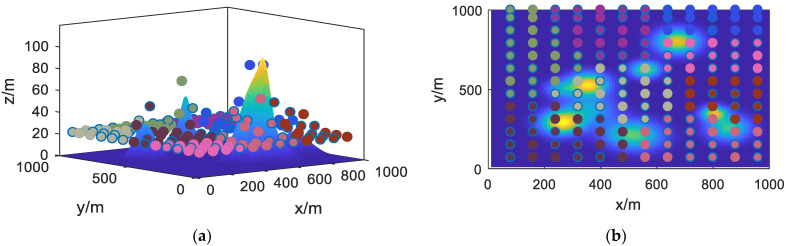
Clustering Algorithm Results Graph. (**a**) 3D view; (**b**) 2D view.

**Figure 14 sensors-23-09122-f014:**
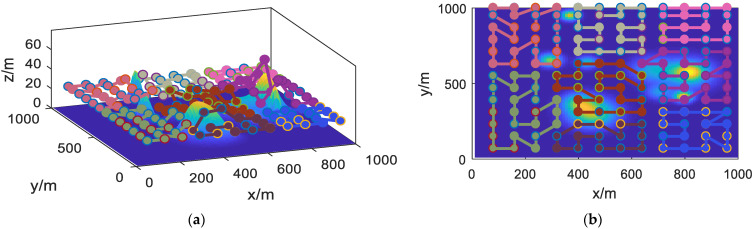
UAV Cluster Task Search Results. (**a**) 3D view; (**b**) 2D view.

**Figure 15 sensors-23-09122-f015:**
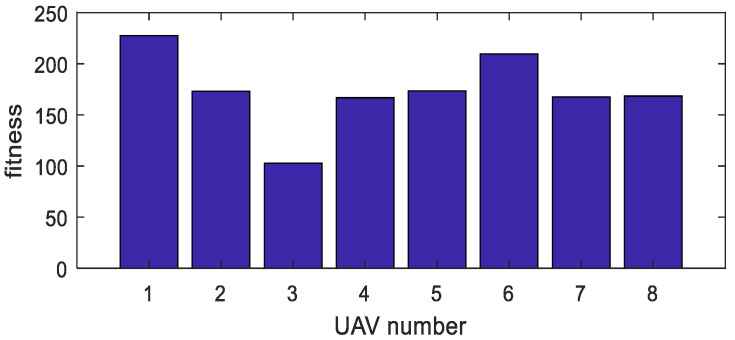
Fitness Values for Each UAV.

**Figure 16 sensors-23-09122-f016:**
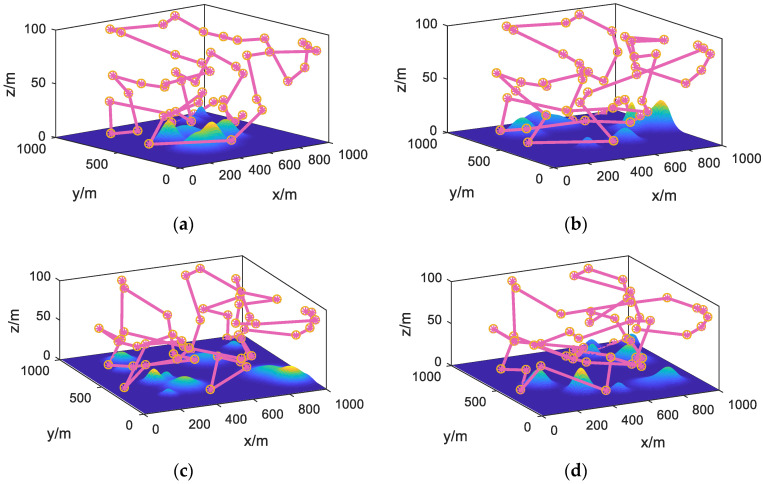
Three-Dimensional Trajectory Planning Results under Different Algorithms (Replacement). (**a**) PSOHAC; (**b**) PSO; (**c**) ACO; (**d**) GA.

**Figure 17 sensors-23-09122-f017:**
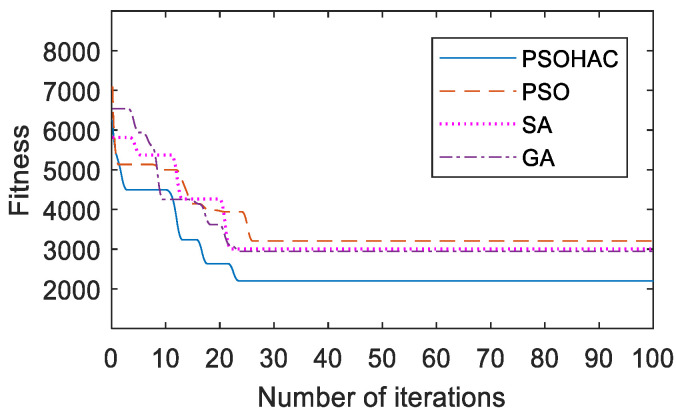
Convergence Curves of Different Algorithms.

**Table 1 sensors-23-09122-t001:** Summary of Existing Research.

Ref	Research Objectives	Method/Algorithm	Number of UAV	Application Scenario
[10]	Improve the efficiency of task allocation	multi-objective optimization	Cluster	2D
[11]	Balanced UAV Task Assignment	Simulated Annealing-Based Strategy	Cluster	3D
[12]	Improve the efficiency of task allocation	Modeling	Cluster	2D
[13]	Path optimization	Improved Harris Hawks Optimization algorithm	Single	3D
[14]	Path optimization Improve search efficiency	Modeling	Cluster	2D
[15]	Path optimization	Hybrid Salp Swarm Algorithm Aquila Optimizer	Single	3D
[16]	Path optimization	Reverse Glowworm Swarm Optimization	Single	3D
[17]	Balanced UAV Task Assignment Improve the efficiency of task allocation	pigeon-inspired fuzzy multi-objective optimization algorithm	Cluster	3D

**Table 2 sensors-23-09122-t002:** Constant Parameter Values and Descriptions in this Study.

Parameter	Descriptions	Value
$\alpha_{s}$	UAV Cone Beam Search Angle	30°
$d_{sfmax}$	UAV Maximum Safety Distance	5 m
$h_{smax}$	UAV Maximum Search Altitude	5 m
$\alpha_{d}$	Energy-to-Distance Ratio Constant	0.8
$v_{i}$	UAV Constant Travel Speed	2.1 m/s
$t_{in}$	Algorithm Initialization Time	5.5
$E_{il}$	Horizontal Unit Distance Energy Consumption	0.2
$E_{iv}$	Vertical Unit Distance Energy Consumption	0.5

**Table 3 sensors-23-09122-t003:** Comparative Experimental Results.

Trajectory Planning Method	Area Division Method	Total Path Length/m	UAV Flight Time/s	Balanced Energy Consumption Efficiency	Overall Energy Consumption Efficiency
Method used in this paper	Method used in this paper	1348.6	78.2	0.712	15.82
	K-means clustering	1346.5	78.9	0.644	17.07
	C-means clustering	1370.4	81.5	0.62	16.81
GA	Method used in this paper	1343.5	79.9	0.604	16.81
	K-means clustering	1351.2	80.4	0.584	16.89
	C-means clustering	1380.6	82.1	0.592	16.82
PSO	Method used in this paper	1345.3	80.1	0.636	16.90
	K-means clustering	1329.0	79.1	0.596	16.75
	C-means clustering	1379.8	82.13	0.5948	16.86
ACO	Method used in this paper	1345.2	80.1	0.636	16.79
	K-means clustering	1341.0	79.8	0.608	16.81
	C-means clustering	1355.3	80.7	0.612	16.77
SA	Method used in this paper	1342.2	79.9	0.6	16.53
	K-means clustering	1333.3	79.3	0.632	16.85
	C-means clustering	1369.6	81.5	0.588	16.84

## Data Availability

The raw/processed data required to reproduce these findings cannot be shared at this time as these data also form part of an ongoing study.

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
