# Peer review of "UAV Cluster Mission Planning Strategy for Area Coverage Tasks"

_sensors, 2023, doi:10.3390/s23229122_

Round 1

Reviewer 1 Report

Dear Authors,

The paper you proposed is very interesting and shows a reasonable attitude towards the UAV cluster search mission planning.

The structure of the paper is correct and allows for easy following of your ideas and performed research. The language applied is clear and shows the achieved results precisely. The conclusions were related to the presented results.

Some mistakes appeared in those results in Table 3. The UAV flight time is around 80 seconds and the distance flown in that time is 1348.6 km, which is in contradiction with the declared flight speed of the simulated UAV. Please, review those elements and check their correctness.

After checking and eventually correcting those data, I believe the paper can be published.

Author Response

According to your opinion, I have changed the incorrect statistical unit in Table 3 from kilometers to meters

Reviewer 2 Report

The authors propose a mission planning strategy for multi-drone clusters in area coverage tasks. The interesting aspect of this study is the use of an improved fuzzy C-clustering algorithm to determine the drone task area together with an optimized ant colony algorithm proposed to plan a drone path. The simulation confirms the proposed method's efficiency in achieving full coverage of the task area.

The content of the paper is interesting. The study subject is relevant. The presentation of the result is acceptable. The abstract reflects the paper's context. Analysis of the problem state is acceptable but should include some recommendations about the areas of exploitation of such strategy. For example, authors can provide this recommendation based on the reviews of drone applications:

- Mukhamediev, R.I., Symagulov, A., et al. Review of some applications of unmanned aerial vehicles technology in the resource-rich country, Applied Sciences (Switzerland), 2021, 11(21), 10171

- Li, Y., Liu, M., Jiang, D., Application of Unmanned Aerial Vehicles in Logistics: A Literature Review, Sustainability, 2022, 14(21), 14473

These recommendations can be introduced in the paper’s introduction or conclusion.

The paper is well illustrated and provides all the necessary information about drone mission planning strategy f in the form of pictures and tables.

And short comments. Could you explain, what is the difference between the conception of “drone cluster” and “drone swarm”?

Author Response

Following your suggestion, I have read and added references 7 and 21. I believe these are two very high-quality literature, and at the end of the first paragraph of the introduction, I have added an explanation of UAV cluster and UAV swarm.

Reviewer 3 Report

In this paper, an efficient mission planning stratgy for UAV clustersin area coverage tasks is proposed to solve these challenges such as uneven task assignment, low task efficiency, and high energy consumption. Furthermore, an optimized particle swarm hybrid ant colony algorithm is proposed to plan the UAV cluster task path. Moreover, some improvements in writing need to be noticed. The detailed comments are given as follows:

1. The innovative points of this article are not clearly expressed, and it is recommended to provide a detailed explanation in the first section.

2. This paper mentions Figure 17, but there is no corresponding image in the text.

3. Due to the lack of Figure 17, the comparative analysis is quite confusing and unclear.

4. The title of the subgraph is missing in Figure 5.

5. The expression of formula (9) in this article is unclear.

None

Author Response

Following your suggestion, we have added Figure 17 and added sub image titles for Figure 5. In addition, we also explained the meaning of formula 9.
